# PROVABLE BENEFITS OF DEEP HIERARCHICAL RL

## ABSTRACT

Modern complex sequential decision-making problem often requires both low-level policy and high-level planning. Deep hierarchical reinforcement learning (Deep HRL) admits multi-layer abstractions which naturally model the policy in a hierarchical manner, and it is believed that deep HRL can reduce the sample complexity compared to the standard RL frameworks. We initiate the study of rigorously characterizing the complexity of Deep HRL. We present a model-based optimistic algorithm which demonstrates that the complexity of learning a near-optimal policy for deep HRL scales with the *sum* of number of states at each abstraction layer whereas standard RL scales with the *product* of number of states at each abstraction layer. Our algorithm achieves this goal by using the fact that distinct high-level states have similar low-level structures, which allows an efficient information exploitation and thus experiences from different high-level state-action pairs can be generalized to unseen state-actions. Overall, our result shows an *exponential* improvement using Deep HRL comparing to standard RL framework.

## 1 INTRODUCTION

Reinforcement learning (RL) is a powerful tool to solve sequential decision making problems in various domains, including computer games (Mnih et al., 2013), Go (Silver et al., 2016), robotics (Schulman et al., 2015). A particular feature in these successful applications of RL is that these tasks are concrete enough to be solved by primitive actions and do not require high-level planning. Indeed, when the problem is complex and requires high-level planning, directly applying RL algorithms cannot solve the problem. An example is the Atari game Montezuma's Revenge, in which the agent needs to find keys, kills monsters, move to correct rooms, etc. This is notoriously difficulty problem that requires more sophisticated high-level planning.

Hierarchical reinforcement learning (HRL) is powerful framework that explicitly incorporate high-level planning. Roughly speaking, HRL divides the decision problem into multiple layers and each layer has its own state space. States in higher layers represent more abstraction and thus higher layer states are some time named meta-states. When number of layers of abstraction is large, we call this framework, *deep hierarchical reinforcement learning* (deep HRL). In deep HRL, the agent makes decisions by looking at states from all layers. The dependency on higher layer states represents high-level planning. HRL has been successfully applied to many domains that require high-level planning, including autonomous driving (Chen et al., 2018), recommendation system (Zhao et al., 2019), robotics (Morimoto & Doya, 2001). Recently, extension to imitation learning has also been studied (Le et al., 2018).

While being a practically powerful framework, theoretical understanding on HRL is still limited. Can we provably show the benefit of using HRL instead of naïve RL? In particular, what can we gain from multi-layer abstraction of deep HRL? Existing theories mostly focus on option RL setting, which transits from the upper layer to lower layer when some stopping criterion is met (Fruit et al., 2017). This is different from our setting requiring horizons in each layer to be the same, which is common in computer games and autonomous driving Moreover, the number of samples needed in Fruit et al. (2017) is proportional to the total number of states and total number of actions. In our setting, both the total number of states and number of actions can be exponentially large and hence their algorithm becomes impractical.

We initiate the study of rigorously characterizing the complexity of deep HRL and explaining its benefits compared to classical RL. We study the most basic form, tabular deep HRL in which there

are total $L$-layers and each layer has its own state space $\mathcal{S}_\ell$ for $\ell \in [L]$. One can simply apply classical RL algorithm on the enlarged state space $\mathcal{S} = \mathcal{S}_1 \times \mathcal{S}_2 \times \cdot \times \mathcal{S}_L$. The sample complexity will however depend on the size of the enlarged states space $|\mathcal{S}| = \Pi_{\ell=1}^L |\mathcal{S}_\ell|$. In this paper, we show because of the hierarchical structure, we can reduce the sample complexity *exponentially*, from $\propto \mathrm{poly}(\Pi_\ell^L |\mathcal{S}_\ell|)$ to $\propto \sum_{\ell=1}^L \mathrm{poly}(|\mathcal{S}_\ell|)$. We achieve this via a model-based algorithm which carefully constructs confidence of the model in a hierarchical manner. We fully exploit the structure that lower-level MDPs of different high-level states share the same transition probability, which allows us to combine the information collected at different high-level states and use it to give an accurate estimator of the model for all low-level MDPs. Due to this information aggregation, we are able to improve the sample complexity bound. To our knowledge, this is the first theoretical result quantifying the complexity of deep HRL and explain its benefit comparing to classical RL.

**Organization.** This paper is organized as follows. In Section 2 we discuss related work. In Section 3, we review basic RL concepts and formalize deep HRL. In Section 4, we present our main algorithm and present its theoretical guarantees. In Section 5, we give a proof sketch of our main theorem. We conclude in Section 6 and defer some technical lemmas to appendix.

## 2 RELATED WORK

We are going to provide several related literatures on tabular MDP and hierarchical reinforcement learning in this section.

As for tabular MDP, many works focus on solving MDP with a simulator which can provide samples of the next state and reward given the current state-action pair. These work includes Lattimore & Hutter (2012); Azar et al. (2013); Sidford et al. (2018b;a); Agarwal et al. (2019). Since we do not need to consider the balance between exploration and exploitation, this setting is easier than the setting of minimizing the regret.

There are also a line of work on analysis of the regret bound in RL setting. Jaksch et al. (2010) and Agrawal & Jia (2017) propose a model-based reinforcement learning algorithm, which estimates the transition model using past samples and add a bonus to the estimation. Their algorithms achieve regret bound $\tilde{\mathcal{O}}(\sqrt{H^4 S^2 AT})$ and $\tilde{\mathcal{O}}(\sqrt{H^3 S^2 AT})$ respectively. Later, the UCBVI algorithm in Azar et al. (2017) adds bonus term to the Q function directly, and achieves the regret bound $\tilde{\mathcal{O}}(\sqrt{H^2 SAT})$, which matches the lower bound when the number of episode is sufficiently large. Adopting the technique of variance reduction, the vUCQ algorithm in Kakade et al. (2018) improves the lower order term in the regret bound. Jin et al. (2018) proposed a model-free Q-learning algorithm is proved to achieve regret bound $\tilde{\mathcal{O}}(\sqrt{H^3 SAT})$.

Hierarchical reinforcement learning Barto & Mahadevan (2003) are also broadly studied in Dayan & Hinton (1993); Parr & Russell (1998); Sutton et al. (1999); Dietterich (2000); Stolle & Precup (2002); Bacon et al. (2017); Florensa et al. (2017); Frans et al. (2017). The option framework, which is studied in Sutton et al. (1999); Precup (2001), is another popular formulation used in hierarchical RL. In Fruit et al. (2017), the regret analysis is carried on option reinforcement learning, but their analysis only applies to the setting of option RL. To our current knowledge, there is no such work analyzing the regret bound of multi-level hierarchical RL.

## 3 PRELIMINARIES

### 3.1 EPISODIC MARKOV DECISION PROCESS

In this paper, we consider finite horizon Markov decision process (MDP). An MDP is specified by a tuple $(\mathcal{S}, \mathcal{A}, H, P, r)$, where $\mathcal{S}$ is the (possibly uncountable) state space, $\mathcal{A}$ is a finite action space, $H \in \mathbb{Z}_+$ is a planning horizon, $P : \mathcal{S} \times \mathcal{A} \to \triangle(\mathcal{S})$ is the transition function, and $r : \mathcal{S} \times \mathcal{A} \to [0, 1]$ is the reward function. At each state $s \in \mathcal{S}$, an agent is able to interact with the MDP by playing an action $a \in \mathcal{A}$. Once an action $a$ is played on state $s$, the agent receives an immediate reward $r(s, a) \in [0, 1]$ [1], and the state transitions to next state $s'$ with probability $P(s'|s, a)$. Starting

---

[1]Here we only consider cases where rewards are in $[0, 1]$, but it is easy to know that this result can generalize to rewards in a different range using standard reduction Sidford et al. (2018a). We can also generalize our

from some initial state $s_1 \in \mathcal{S}$ (draw from some distribution), the agent is able to play $H$ steps (an episode) and then the system resets to another initial state $s_1$ sampled from the initial distribution.

For an MDP, our goal is to obtain an *optimal* (will be precise shortly) policy, $\pi : \mathcal{S} \to \mathcal{A}$, which is a function that maps each state to an action. If an agent always follows the action given by a policy $\pi$, then it induces a random trajectory for an episode: $s_1, a_1, r_1, s_2, a_2, r_2, \ldots, s_H, a_H, r_H$ where $r_1 = r(s_1, a_1)$, $s_2 \sim P(\cdot|s_1, a_1)$, $a_2 \sim \pi(s_2)$, etc. The value function and $Q$-function at step $h$ of a given policy is defined as $V_h^\pi(s) = \mathbf{E}_\pi \left[ \sum_{h'=h}^H r_{h'} \big| s_h = s \right]$ and $Q_h^\pi(s, a) = \mathbf{E}_\pi \left[ \sum_{h'=h}^H \big| s_h = s, a_h = a \right]$, where the expectation is over all sample trajectories. Then the optimal policy, $\pi^*$, is defined to be the policy with largest $V_1^\pi(s_1)$ for all $s_1 \in \mathcal{S}$. For any optimal policy, its value and $Q$-function satisfy the following Bellman equation

$$\forall s \in \mathcal{S}, a \in \mathcal{A}, h \in [H] : \quad Q_h^*(s, a) = r(s, a) + \mathbf{E}_{s' \sim P(\cdot|s, a)} V_{h+1}^*(s'), \quad V_h^*(s) = \max_{a \in \mathcal{A}} Q_h^*(s, a)$$

$$\text{and} \quad V_{H+1}^*(s) = 0. \tag{1}$$

We consider the MDP problem in the online learning setting, where the probability transition is unknown. However, our goal is still to collect the maximum amount reward, i.e., play a policy that is comparable to the optimal one. Therefore, the agent needs to learn to play through trial and error, i.e., improving the policy by learning from experiences. Suppose we allow the agent to play in total $K \geq 1$ episodes. For each episode, the agent is following a policy $\pi^k$, which is computed based on her experiences collected from episodes $1, 2, \ldots, k-1$. To measure the performance of the agent, we use the following standard regret formulation, which compares the reward collected by the agent to the performance of an optimal policy.

$$R(K) = \sum_{k=1}^K \left( V_1^*(s_1) - V_1^{\pi_k}(s_1) \right). \tag{2}$$

Note that, if the agent learns nothing, we then expect $R(K) \propto K$. But if the agent is able to learn, then the average regret, $R(K)/K$, which measures the average error per step, goes to 0 when $K$ becomes large. In the online MDP literature, model based algorithms (e.g. Jaksch et al. (2010)) achieves regret $R(K) \leq \tilde{O}\left( \sqrt{H^2 |\mathcal{S}|^2 |\mathcal{A}| H K} \right)$.

## 3.2 DEEP HIERARCHICAL MDP

In this section we introduce a special type of episodic MDPs, the hierarchical MDP (hMDP). If we view them as just normal MDPs, then their state space size can be exponentially large. Formally, each hMDP consists of $L$ levels of episodic MDPs with the $\ell$-th level having planning horizon $H_\ell$. One can view the $\ell$-th level MDP as a subtask of the $(\ell+1)$-th level MDP. To transition between two state in $(\ell+1)$-th level, the agent needs to play an episode in $\ell$-th level MDP (the state transition will be defined formally shortly). Therefore, the total planning horizon of the hierarchical MDP is $H = \prod_{i=1}^L H_\ell$.

For each step $h$ in the hMDP, we can represent it by a tuple $(h_1, \cdots, h_L)$, where $h_\ell \in [H_\ell]$ is the step of $\ell$-th level MDP for $1 \leq \ell \leq L$. We use $\tilde{H}_\ell = \prod_{i=\ell}^L H_i$ to denote the effective horizon of level $\ell$, which represents the total number of actions in $\mathcal{A}_\ell \cup \cdots \cup \mathcal{A}_L$ needed to be played in an episode of the full hMDP. Note that, for each $h = (h_1, \cdots, h_L) \in [1, H]$, we have $h + 1 = (h'_1, \cdots, h'_L)$ is the immediate next lexicographical tuple of $(h_1, \cdots, h_L)$.

We now describe how the agent can interact with the full hMDP. In fact, in each step $h$, only an action in one level can be played. This level is given by the function $\sigma : [H] \to [L]$, formally defined as

$$\sigma(h) = \arg\max_{\ell \in [L]} \{h_\ell = H_\ell\} + 1.$$

It characterizes the lowest level of MPDs which does not reach the last step in its horizon.

---

result to the setting where the reward is stochastic, since estimating the reward accurately requires much fewer samples than estimating the transition.

Figure 1: Demonstration of Hierarchical MDP: 3 levels of MDP, level 1 (lowest level) has 3 states, level 2 has 3 states, level 3 (highest level) has 4 states, the total number of states is 36.

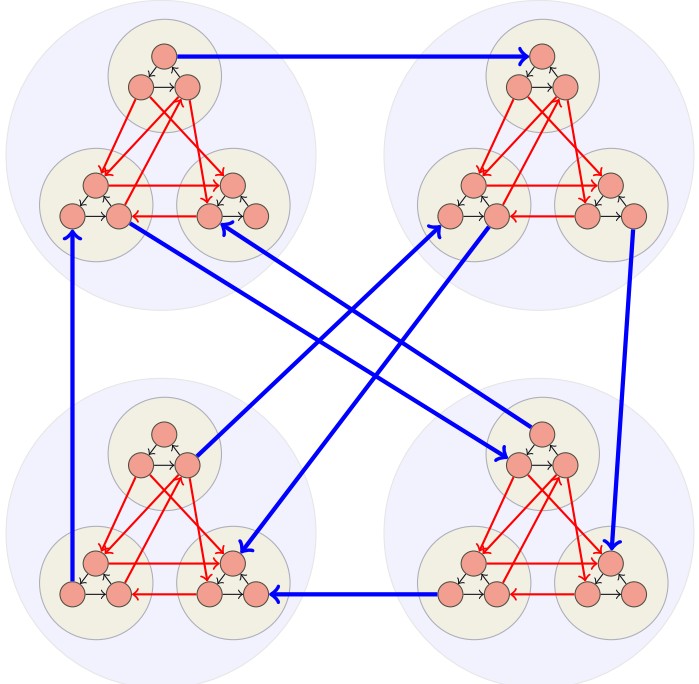

To make it formal for the state-transition, we use $\mathcal{S}_\ell$ to denote the set of states at level $\ell$, $\mathcal{A}_\ell$ to denote the set of actions at level $\ell$. To be convenient, we assume for every $\ell \in [L]$, for any $H_\ell$-length trajectory in the $\ell$-th level MDP, the last state always falls in $\mathcal{S}_\ell' \subset \mathcal{S}_l$, which we call as the *endstates* in level $\ell$. At step $h$ of the full hMDP, the full state is described as a length-$L$ tuple: $(s_1, \cdots, s_L)$. For any $1 \le \ell < \sigma(h)$, we immediately have $s_\ell \in \mathcal{S}_\ell'$ is an endstate of level $\ell$. Note that the total number of states of the full MDP is $\prod_{\ell=1}^L |\mathcal{S}_\ell|$, which is exponentially larger than the average size of a level's state space.

Now we define the transition. At step $h$ of the full hMDP, the agent plays an action $a_h^{\sigma(h)} \in \mathcal{A}_{\sigma(h)}$ for the $\sigma(h)$-level MDP. The state of this MDP triggers a transition at level $\sigma(h)$:

$$s_{h+1}^{\sigma(h)} \sim P\left( \cdot \mid s_h^{\sigma(h)}, a_h^{\sigma(h)}, s_h^{\sigma(h)-1} \right)$$

Note that the probability transition is determined by the state-action-ending-state tuple $(s_h^{\sigma(h)}, a_h^{\sigma(h)}, s_h^{\sigma(h)-1})$, instead of single state-action pair. Moreover, all the MDPs with level lower than $\sigma(h)$ will reset their state based on some initial distribution $P_0^i$:

$$s_{h+1}^i \sim P_0^i(\cdot), \quad \forall 1 \le i \le \sigma(h) - 1,$$

and all the MDPs with level higher than $\sigma(h)$ will keep their states unmoved.

For any given $\ell \in [L]$, we use $\mathcal{E}_\ell$ to denote the state-action-ending-state tuple at level $\ell$:

$$\mathcal{E}_\ell = \{(s^\ell, a^\ell, s^{\ell-1}) \mid s^\ell \in \mathcal{S}_\ell, \ a^\ell \in \mathcal{A}_\ell, \ s^{\ell-1} \in S_{\ell-1}'\}.$$

As for the reward, we use $r(s_h^1, \cdots, s_h^L, a_h^{\sigma(h)}) \in [0, 1]$ to denote the immediate reward obtained after executing the action $a_h^{\sigma(h)}$. We illustrate the hierarchical MDP model in Figure 1.

**An Example: Autonomous Driving.** We here give a more concrete example. Suppose we want our vehicle to reach the destination, while not hitting obstacles or crashing into another vehicles or pedestrians. We use the following hierarchical MDP structure to formulate this problem.

Level 1 represents the status (e.g. position on the road, whether has an obstacle in the front) of the vehicle, and the ending state represents whether the vehicle avoids all the obstacles, other vehicles, pedestrians and arrives at the end of the road. Level 2 represents the road map, and the ending state represents whether the vehicle reaches the desired position.

At each time step, if the vehicle does not reach the end state of level 1, that is, it still on the road and not at a crossing, then the vehicle needs to decide whether speeding up, slowing down or dodging the obstacle in the front. If the vehicle reaches the end state of level 1, that is, it arrives at the end of a road, then it needs to decide whether going straight forward, turning left or turning right. This process ends if and only if the vehicle reaches the desired position.

### 3.3 DEEP HIERARCHICAL REINFORCEMENT LEARNING OBJECTIVE

Suppose the environment is an hMDP. The hierarchical structure and the reward is known but the transition models are not known. Similar to the classic RL setting, our agent needs to interact with the unknown hMDP while being able to accumulate the amount of rewards comparable to an optimal policy. Our goal is design an algorithm that minimizes the regret defined in Equation (2).

Since an hMDP is very special compared to a normal MDP, we redefine its related quantities here. The policy $\pi$ is a mapping from $\mathcal{S}_1 \times \cdots \times \mathcal{S}_L \times [H]$ to $\mathcal{A}_1 \cup \cdots \mathcal{A}_L$, where $\pi(s^1, \cdots, s^L, h) \in \mathcal{A}_\ell$ if and only if $\sigma(h) = \ell, s^1 \in \mathcal{S}'_1, \cdots, s^{\ell-1} \in \mathcal{S}'_{\ell-1}$ and $s^\ell \in \mathcal{S}_\ell, \cdots, s^L \in \mathcal{S}_L$. Given a policy $\pi$, and step $h$, the value function and $Q$ function are again defined in Equation (1), but can be rewritten as,

$$Q_h^\pi(s^1, \cdots, s^L; a) = r(s^1, \cdots, s^\ell; a)$$
$$+ \mathbf{E}_{\tilde{s}^1 \sim P_0^1, \cdots, \tilde{s}^{\ell-1} \sim P^{\ell-1}, \tilde{s}^\ell \sim P(\cdot|e)} V_{h+1}^\pi(\tilde{s}^1, \cdots, \tilde{s}^\ell, s^{\ell+1} \cdots, s^L),$$
$$V_h^\pi(s^1, \cdots, s^L) = Q_h^k(s^1, \cdots, s^L; \pi(s^1, \cdots, s^L, h))$$
$$V_{H+1}^\pi(s^1, \cdots, s^L) = 0, \quad \forall s^\ell \in \mathcal{S}_\ell, 1 \le \ell \le L.$$

Our objective is to find a policy $\pi^*$ such that the value function $V_h^\pi$ is maximized for all states and steps in a horizon. We use $V_h^*$ and $Q_h^*$ to denote the optimal value function and optimal Q-function, which is the value function and the Q-function when applying the optimal policy $\pi^*$.

## 4 ALGORITHM

In this section, we will present a model-based hierarchical reinforcement learning algorithm, together with its regret bound analysis.

### 4.1 MODEL-BASED HIERARCHICAL REINFORCEMENT LEARNING ALGORITHM

To formally present our algorithm, we first explain the high-level ideas. Note that the full model size is $\mathcal{O}\left(\prod_{\ell=1}^L |\mathcal{S}_\ell||\mathcal{E}_\ell|\right)$, where $|\mathcal{S}_\ell|$ is the number of states in level $\ell$ and $|\mathcal{E}_\ell|$ is the number of state-action-endstate tuples in level $\ell$. However, we notice that there are rich structures for the algorithm to exploit: low-level MDPs corresponding to different high-level states share the same transition model. Recall that, our eventual goal to learn the hierarchical model with number of samples much less than $\prod_{\ell=1}^L |\mathcal{S}_\ell||\mathcal{E}_\ell|$. To achieve this goal, we group our samples obtained from transition models by state-action-endstate tuple, and samples obtained from initial distributions by levels: even if the samples are collected at a different high-level state-action pair, they are grouped to a same set as long as they come from a same state-action-endstate pair. We then use the samples from a level to estimate the MDP initial distribution corresponding to that level. In effect, to estimate all the MDPs corresponding to a level accurately, we only need to visit this level a number of times proportional to the size of a *single* MDP in this level, which is far smaller than the model of all MDPs in this level combined.

Next, we explain how we can deploy the algorithm in the online setting, where we can only visit a state by following the trajectory of some policy. Initially, we have no knowledge of the MDPs corresponding to each level, we just initialize each of them to an arbitrary MDP. Suppose we play the whole game for $K$ episodes, and for each episode, we play $H = \prod_{\ell=1}^L H_\ell$ steps, where $H_\ell$

is the horizon of an MDP in level $\ell$. Suppose at episode $k \in [K]$ and step $h \in [H]$, the transition happens at level $\ell$ (which means $\sigma(h) = \ell$). We denote the full state we observed as $(s_{h,k}^1, \cdots, s_{h,k}^L)$ and the action we take as $a_{h,k}^\ell$. Then we collect data samples of the form $(e_{h,k}^\ell, s_{h+1,k}^\ell)$, where $e_{h,k}^\ell = (s_{h,k}^\ell, a_{h,k}^\ell, s_{m,k}^{\ell-1}) \in \mathcal{E}_\ell$ is a state-action-endstate tuple at level $\ell$, and also data samples of the form $(i, s_{h+1,k}^i)$ for every $1 \leq i \leq \ell-1$. We add them to a buffer, $s_{h+1,k}^\ell$ to $N_{e_{h,k}}$ corresponding to state-action-endstate tuple $e_{h,k}$, and $s_{h+1,k}^i$ to $M_i$ corresponding to level $i$. Here $M_i$ and $N_{e_{h,k}}$ are multisets, i.e., their elements can be repeated. We use $N_{e_{h,k}}$ to estimate probability transition $P(\cdot|e_{h,k}^\ell)$, and $M_i$ to estimate $P_0^i(\cdot)$ respectively.

When a new episode $k$ starts, we first do a model estimation based on all the samples collected and partitioned. Using this estimated model, a value function and a Q-function is computed. However, the estimation error always exists in the estimated model due to insufficient samples from certain state-action-endstate tuples. To account for this, we estimate the model uncertainty based on concentration inequalities. Specifically, we add the uncertainties to our value function estimator, and use the modified value function to play for the new episode. Note that doing so encourages exploration on unexperienced state-action pairs. In fact, as we will show shortly, by using appropriate uncertainty estimation, the model becomes more accurate if the algorithm makes a mistake (e.g., by playing a less-optimal action). With a pigeon hole principle argument, we can show that our algorithm achieves a low regret bound.

Our model-based algorithm is formally presented in Algorithm 1. We denote our estimator of the initial distribution at level $\ell$ as

$$\tilde{P}_{k,0}^\ell(s) = \frac{\#\{s \in M_\ell\}}{|M_\ell|}, \tag{3}$$

where $\#\{s \in M_\ell\}$ and $|M_\ell|$ are the number of appearance of state $s$ in buffer $M_\ell$ and the total number of states in buffer $M_\ell$, respectively. We also denote our estimator of transition distribution at state-action-endstate tuple $e$ as

$$\tilde{P}_k(s|e) = \frac{\#\{s \in N_e\}}{|N_e|}, \tag{4}$$

where $\#\{s \in N_e\}$ and $|N_e|$ are the number of appearance of state $s$ in buffer $N_e$ and the total number of states in buffer $N_e$, respectively. With this these estimators, we use dynamic programming to solve for the $Q$-function and value functions as follows,

$$Q_h^k(s^1, \cdots, s^L; a) = r(s^1, \cdots, s^L; a) + b(k, h, \ell, e)$$
$$+ \mathbf{E}_{\tilde{s}^1 \sim \tilde{P}_{k,0}^1, \cdots, \tilde{s}^{\ell-1} \sim \tilde{P}_{k,0}^{\ell-1}, \tilde{s}^\ell \sim \tilde{P}_k(\cdot|e)} V_{h+1}^k(\tilde{s}^1, \cdots, \tilde{s}^\ell, s^{\ell+1} \cdots, s^L), \tag{5}$$
$$V_h^k(s^1, \cdots, s^L) = \min\left\{H, \max_{a \in \mathcal{S}_\ell}\left[Q_h^k(s^1, \cdots, s^L; a)\right]\right\},$$

where for $1 \leq h \leq H$, $\ell = \sigma(h)$, $e \in \mathcal{E}_\ell$ and $V_{H+1}^k(s^1, \cdots, s^L) = 0$. Here the bonus function $b(k, h, \ell, e)$ is used to estimate the uncertainty of the $Q, V$ estimator, which are defined as follows:

$$b(k, h, \ell, e) = H \min\left\{1, \sqrt{\frac{8(|\mathcal{S}_\ell| + \log(4L^2|\mathcal{S}_\ell||\mathcal{E}_\ell|k^2/\delta))}{n(k-1, e)}}\right\}$$
$$+ H \sum_{i=1}^{\ell-1} \min\left\{1, \sqrt{\frac{8(|\mathcal{S}_i| + \log(4L^2|\mathcal{S}_i||\mathcal{E}_\ell|k^2/\delta))}{(k-1)\tilde{H}_i}}\right\}, \tag{6}$$

where $\delta$ is a constant between $[0, 1]$ to be specified before, and $n(k-1, e)$ is the number of times we encountered state-action-endstate tuple before $k$-horizon. These bonus functions bound the difference between the estimated $Q$-functions to the exact value (per-step) with high probability. For episode $k$, our exploratory policy is then

$$\pi^k(s^1, \cdots, s^L, h) = \arg\max_{a \in \mathcal{A}_\ell}\left[Q_h^k(s^1, \cdots, s^L; a)\right]. \tag{7}$$

---

**Algorithm 1** Model-based Algorithm for Hierarchical RL

---

1: **Input:** An MDP with hierarchical structure, $\delta$
2: **Initialize:** $M_\ell = N_e = \varnothing$ (elements repeatable) for every $1 \le \ell \le L, e \in \mathcal{E}_\ell$;
3: **Initialize:** $b(k, h, \ell, e)$ as in formula (6)
4: **for** $k = 1 : K$ **do**
5:     Calculate $V_h^k, Q_h^k, \pi_k$ with uses formula (5), (7).
6:     **for** $h = 1 : H$ **do**
7:         Play action $a_{h,k}^{\sigma(h)} = \arg\max_{a \in \mathcal{A}_{\sigma(h)}} \left[ Q_h^k(s_{h,k}^1, \cdots, s_{h,k}^L; a) \right]$;
8:         Get next state $(s_{h+1,k}^1, \cdots, s_{h+1,k}^L)$;
9:         **for** $i = 1 : \sigma(h)$ **do**
10:             Put $s_{h+1,k}^i$ into $M_\ell$;
11:         Put $s_{h+1,k}^\ell$ into $N_{(s_{h,k}^\ell, a_{h,k}^\ell, s_{h,k}^{\ell-1})}$.
12:     Update
$$\tilde{P}_{k,0}^\ell(s) = \frac{\#\{s \in M_\ell\}}{|M_\ell|}, \quad \forall 1 \le \ell \le L, s \in \mathcal{S}_\ell,$$
$$\tilde{P}_k(s|e) = \frac{\#\{s \in N_e\}}{|N_e|}, \quad \forall 1 \le \ell \le L, e \in \mathcal{E}_\ell, s \in \mathcal{S}_\ell.$$

---

## 4.2 REGRET BOUND FOR ALGORITHM 1

In this subsection we provide a formal guarantee for Algorithm 1. We present a proof sketch in the next section, and the full proof is deferred to appendix.

**Theorem 4.1.** *Suppose we run Algorithm 1 for $K \ge 1$ episodes on an hMDP. For $k \in [K]$, let $\pi^k$ be policy played by the algorithm in episode $k$. Then we have, with probability at least $1 - \delta$,*

$$R(K) = \sum_{\ell=1}^{L} \tilde{O} \left( H\tilde{H}_\ell |\mathcal{E}_\ell| + H\sqrt{K\tilde{H}_\ell \cdot |\mathcal{E}_\ell|(|\mathcal{S}_\ell| + \log \delta^{-1})} \right).$$

*where $\delta \in (0,1)$ and $R(K)$ is defined in Equation (2).*

From this theorem, we observe that the regret bound only depends on $\sum_{\ell=1}^{L} \sqrt{|\mathcal{S}_\ell||\mathcal{E}_\ell|}$, where $|\mathcal{E}_\ell| = |\mathcal{S}_\ell||\mathcal{A}_\ell||\mathcal{S}_{\ell-1}'|$ (here $|\mathcal{E}_{\ell-1}'|$ is the number of endstates at level $\ell - 1$). Usually, the number of actions and the number of endstates at a level are much smaller than the number of states and can be viewed as constant. In this way our regret bound only depends on $\sum_{\ell=1}^{L} |\mathcal{S}_\ell|$. It means after $K = \tilde{\Omega} \left( \sum_{\ell=1}^{L} H\sqrt{\tilde{H}_\ell |\mathcal{E}_\ell||\mathcal{S}_\ell|} \right)$ episodes, the algorithms achieves a constant average regret $R(K)/K = O(1)$ (this is when the agent learns a meaningful amount of information). Let us consider the full hMDP, whose state space size is $\prod_{\ell=1}^{L} |\mathcal{S}_\ell|$. With a model based or model free algorithm like Jaksch et al. (2010); Jin et al. (2019), the number of episodes needed would be $K \gtrsim \prod_{\ell=1}^{L} |\mathcal{S}_\ell|$ to achieve a constant average regret. Note that $\prod_{\ell=1}^{L} |\mathcal{S}_\ell|$ can be exponentially larger than $\text{poly}(\sum_{\ell=1}^{L} |S_\ell|)$, therefore our algorithm achieves an exponential saving in the sample complexity for RL.

## 5 PROOF SKETCH

The proof of Theorem 4.1 is divided into two parts. In the first part, we prove that with high probability, the difference between empirical expectation and true expectation of the value function can be bounded by the bonus $b$. The proof of this property involves estimation of the total variation (TV) distance between a distribution on the state space $\mathcal{S}_\ell$ of level $\ell$ and its empirical estimation using $n$ samples. This TV distance can be bounded by $\tilde{\mathcal{O}}(\sqrt{|\mathcal{S}_\ell|/n})$ with high probability.

The second part of proof tells that if the difference between empirical expectation and true expectation of the value function can be bounded by the bonus $b$, then the estimator $Q_h^k$ of Q function

is always an optimistic estimation to the true Q-function with high probability. That is, for every $s^i \in \mathcal{S}_i$, we have $Q_h^k(s^1, \cdots, s^L) \geq Q_h^*(s^1, \cdots, s^L)$. Then we can show that the regret can be upper bounded by the sum of all bonuses along the sample path. Hence we can obtain the regret bound by summing over all bonuses in each step and horizon. We notice that at level $\ell$ there are only $|\mathcal{E}_\ell|$ distributions we need to estimate, and each one is a distribution on $\mathcal{S}_\ell$. Therefore applying Hölder inequality we obtain the regret bound $\sum_{\ell=1}^L \tilde{\mathcal{O}}(\sqrt{|\mathcal{S}_\ell||\mathcal{E}_\ell|K})$, where we put the dependence on $H$ and $\delta$ into $\tilde{\mathcal{O}}$.

## 6 CONCLUSION

In this paper we prove the benefit of hierarchical reinforcement learning theoretically. We propose a model-based hierarchical RL algorithm which achieves a regret bound that is exponentially better than the naive RL algrorithm. To our knowledge, this is the first theoretical result demonstrating the benefit of using deep hierarchical reinforcement learning. Below we list two future directions.

**Deep Hierarchical Reenforcement Learning with Function Approximation**   The current work focuses the most basic formulation, tabular RL: When state space is large, function approximation is required for generalization across states. Recently, a line of work gave provably polynomial sample complexity upper bound for RL with function approximation under various assumptions (Wen & Van Roy, 2013; Du et al., 2019; Jiang et al., 2017; Yang & Wang, 2019; Jin et al., 2019). An interesting direction is to combine our analysis with these results and obtain guarantees on deep HRL with function approximation.

**Deep Hierarchical Reenforcement Imitation Learning**   Imitation learning is another paradigm where expert's trajectories are available to the agent. Le et al. (2018) presented a framework to combine hierarchical learning and imitation learning. However, there is no formal statistical guarantee. We believe our analysis can be leveraged to understand deep hierarchical imitation learning too.

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

# A    PROOF OF THEOREM 4.1

## A.1    NOTATIONS FOR THE PROOF

We will specify some useful notations in the proof first.

To be convenient, we use $V_h[K], Q_h[K, a]$ to denote $V_h(s_{k,h_1}^1, \cdots, s_{k,h_L}^L)$ and $Q_h(s_{k,h}^1, \cdots, s_{k,h}^L; a)$. We use $V_h[k]^{\otimes \ell}$ to denote the $|\mathcal{S}_1| \times \cdots \times |\mathcal{S}_\ell|$-dimensional tensor whose $(x_1, \cdots, x_\ell)$-element is $V_h(x_1, \cdots, x_\ell, s_{k,h+1}^{\ell+1}, \cdots, s_{k,h+1}^L)$. Given probability distribution $P^i(\cdot)$ $(1 \le i \le \ell)$ over the state space $\mathcal{S}_i$, we use $\mathcal{P}^i$ to denote the operator over tensors:

$$\mathcal{P}^i \left[ V_h^{\otimes \ell}[k] \right] = \sum_{s_i \in \mathcal{S}} P^i(s_i) V_{h_1, \cdots, h_L}(\cdot, \cdots, s_i, \cdots, \cdot, x_\ell, s_{k,h+1}^{\ell+1}, \cdots, s_{k,h+1}^L), \tag{8}$$

which can be understood as taking the expectation on the $i$-th element. We can also define the tensor product operator $\mathcal{P}^{i_1} \otimes \mathcal{P}^{i_2}, \cdots \otimes \mathcal{P}^{i_j}$ for different $i_1, \cdots, i_j$ as the composite operator, where each $\mathcal{P}^{i_j}$ is a operator of dimension $|S_{i_j}|$.

## A.2    PROOF OF THEOREM 4.1

We present the proof of Theorem 4.1 in this subsection. In the next we use $e_{k,h}$ to denote the state-action-endstate pair $(s_{k,h}^l, a_{k,h}^l, s_{k,h}^{l-1})$.

We first present a lemma indicating that with high probability, the difference between the empirical expectation of the value function is bounded by the bonus.

**Lemma A.1.** *With probability at least* $1 - \delta$, *we have*

$$\left| \left[ \tilde{\mathcal{P}}_{k,0}^1 \otimes \cdots \otimes \tilde{\mathcal{P}}_{k,0}^{\ell-1} \otimes \tilde{\mathcal{P}}_k(\cdot|e) - \mathcal{P}_{k,0}^1 \otimes \cdots \mathcal{P}_{k,0}^{\ell-1} \otimes \mathcal{P}_k(\cdot|e) \right] V_{h+1}^k[k]^{\otimes \ell} \right| \le b(k, h, \ell, e)$$

*for every* $k \ge 1, 1 \le h \le H, 1 \le \ell \le L$ *and* $e \in \mathcal{E}_\ell$.

*Proof.* Given any $k \ge 1, 1 \le h \le H, 1 \le \ell \le L$ and $e = (s^l, a^l, s^{l-1}) \in \mathcal{E}_\ell$, we have the following estimation of error between estimated transition model and true transition model.

$$\left| \left[ \tilde{\mathcal{P}}_{k,0}^1 \otimes \cdots \otimes \tilde{\mathcal{P}}_{k,0}^{\ell-1} \otimes \tilde{\mathcal{P}}_k(\cdot|e) - \mathcal{P}_{k,0}^1 \otimes \cdots \mathcal{P}_{k,0}^{\ell-1} \otimes \mathcal{P}_k(\cdot|e) \right] V_{h+1}^k[k]^{\otimes \ell} \right|$$

$$\le \sum_{i=1}^{\ell-1} \left| \left[ \tilde{\mathcal{P}}_{k,0}^1 \otimes \cdots \tilde{\mathcal{P}}_{k,0}^i \otimes \cdots \otimes \mathcal{P}_{k,0}^{\ell-1} \otimes \tilde{\mathcal{P}}_k(\cdot|e) - \tilde{\mathcal{P}}_{k,0}^1 \otimes \cdots \mathcal{P}_{k,0}^i \cdots \mathcal{P}_{k,0}^{\ell-1} \otimes \tilde{\mathcal{P}}_k(\cdot|e) \right] V_{h+1}^k[k]^{\otimes \ell} \right|$$

$$+ \left| \left[ \mathcal{P}_{k,0}^1 \otimes \cdots \mathcal{P}_{k,0}^{\ell-1} \otimes \tilde{\mathcal{P}}_k(\cdot|e) - \mathcal{P}_{k,0}^1 \otimes \cdots \mathcal{P}_{k,0}^{\ell-1} \otimes \mathcal{P}_k(\cdot|e) \right] V_{h+1}^k[k]^{\otimes \ell} \right|$$

$$= \sum_{i=1}^{\ell-1} \left| \left[ \left[ \tilde{\mathcal{P}}_{k,0}^i - \mathcal{P}_{k,0}^i \right] \otimes \tilde{\mathcal{P}}_{k,0}^1 \otimes \cdots \tilde{\mathcal{P}}_{k,0}^{i-1} \otimes \mathcal{P}_{k,0}^{i+1} \cdots \otimes \mathcal{P}_{k,0}^{\ell-1} \otimes \tilde{\mathcal{P}}_k(\cdot|e) \right] V_{h+1}^k[k]^{\otimes \ell} \right|$$

$$+ \left| \left[ \left[ \tilde{\mathcal{P}}_k(\cdot|e) - \mathcal{P}_k(\cdot|e) \right] \otimes \mathcal{P}_{k,0}^1 \otimes \cdots \mathcal{P}_{k,0}^{\ell-1} - \mathcal{P}_{k,0}^1 \otimes \cdots \mathcal{P}_{k,0}^{\ell-1} \right] V_{h+1}^k[k]^{\otimes \ell} \right|$$

$$\le \sum_{i=1}^{\ell-1} \left\| \tilde{\mathcal{P}}_{k,0}^i - \mathcal{P}_{k,0}^i \right\|_1 \left\| \left[ \tilde{\mathcal{P}}_{k,0}^1 \otimes \cdots \tilde{\mathcal{P}}_{k,0}^{i-1} \otimes \mathcal{P}_{k,0}^{i+1} \cdots \otimes \mathcal{P}_{k,0}^{\ell-1} \otimes \tilde{\mathcal{P}}_k(\cdot|e) \right] V_{h+1}^k[k]^{\otimes \ell} \right\|_\infty$$

$$+ \left\| \tilde{\mathcal{P}}_k(\cdot|e) - \mathcal{P}_k(\cdot|e) \right\|_1 \left\| \left[ \mathcal{P}_{k,0}^1 \otimes \cdots \mathcal{P}_{k,0}^{\ell-1} - \mathcal{P}_{k,0}^1 \otimes \cdots \mathcal{P}_{k,0}^{\ell-1} \right] V_{h+1}^k[k]^{\otimes \ell} \right\|_\infty$$

$$\le \sum_{i=1}^{\ell-1} \left\| \tilde{P}_{k,0}^i - P_{k,0}^i \right\|_1 \cdot H + \left\| \tilde{P}_k(\cdot|e) - P_k(\cdot|e) \right\|_1 \cdot H$$

According to Theorem 2.1 in Weissman et al. (2003), for any $1 \le i \le \ell - 1$, with probability at least $1 - \delta$ we have

$$\left\| \tilde{P}_{k,0}^i - P_{k,0}^i \right\|_1 \le \sqrt{\frac{8(|\mathcal{S}_i| + \log \delta^{-1})}{(k-1)\tilde{H}_i}}, \tag{9}$$

where we use the fact that till $k$ horizons, we have $k\tilde{H}_i$ samples for the initial distribution of level $i$. Similarly, with probability at least $1 - \delta$ we have

$$\left\|\tilde{P}_k(\cdot|e) - P_k(\cdot|e)\right\|_1 \leq \sqrt{\frac{8(|\mathcal{S}_\ell| + \log \delta^{-1})}{n(k-1,e)}},$$

where we use $n(k-1,e)$ to denote the number of appearance of state-action-endstate tuple till $k-1$ horizons (including $k-1$ horizon). Replacing $\delta$ with $\delta/(4L^2|\mathcal{S}_i||\mathcal{E}_\ell|k^2)$ and applying union bound on all $1 \leq i \leq \ell$, we obtain

$$\sum_{i=1}^{\ell-1} \left\|\tilde{P}_{k,0}^i - P_{k,0}^i\right\|_1 \cdot H + \left\|\tilde{P}_k(\cdot|e) - P_k(\cdot|e)\right\|_1 \cdot H$$

$$\leq H \left(\sqrt{\frac{8(|\mathcal{S}_\ell| + \log(4L^2|\mathcal{S}_i||\mathcal{E}_\ell|k^2/\delta))}{n(k-1,e)}}\right\} + H \sum_{i=1}^{\ell-1} \min\left\{1, \sqrt{\frac{8(|\mathcal{S}_i| + \log(4L^2|\mathcal{S}_i||\mathcal{E}_\ell|k^2/\delta))}{(k-1)\tilde{H}_i}}\right\}\right)$$

with probability at least $1 - \delta/(4L|\mathcal{S}_i||\mathcal{E}_\ell|k^2)$. Therefore, noticing that

$$\left|\left[\tilde{\mathcal{P}}_{k,0}^1 \otimes \cdots \otimes \tilde{\mathcal{P}}_{k,0}^{\ell-1} \otimes \tilde{\mathcal{P}}_k(\cdot|e) - \mathcal{P}_{k,0}^1 \otimes \cdots \mathcal{P}_{k,0}^{\ell-1} \otimes \mathcal{P}_k(\cdot|e)\right] V_{h+1}^k[k]^{\otimes \ell}\right| \leq H$$

if we choose

$$b(k,h,\ell,e) = H\min\left\{1, \sqrt{\frac{8(|\mathcal{S}_\ell| + \log(4L^2|\mathcal{S}_i||\mathcal{E}_\ell|k^2/\delta))}{n(k-1,e)}}\right\}$$

$$+ H\sum_{i=1}^{\ell-1} \min\left\{1, \sqrt{\frac{8(|\mathcal{S}_i| + \log(4L^2|\mathcal{S}_i||\mathcal{E}_\ell|k^2/\delta))}{(k-1)\tilde{H}_i}}\right\}, \tag{10}$$

then we have

$$\left|\left[\tilde{\mathcal{P}}_{k,0}^1 \otimes \cdots \otimes \tilde{\mathcal{P}}_{k,0}^{\ell-1} \otimes \tilde{\mathcal{P}}_k(\cdot|e) - \mathcal{P}_{k,0}^1 \otimes \cdots \mathcal{P}_{k,0}^{\ell-1} \otimes \mathcal{P}_k(\cdot|e)\right] V_{h+1}^k[k]^{\otimes \ell}\right| \leq b(k,h,\ell,e)$$

with probability at least $1 - \delta/(2L|\mathcal{E}_\ell|k^2)$.

Finally, we apply the union bound on all $k \geq 1$, $1 \leq \ell \leq L$ and $e \in \mathcal{E}_\ell$,

$$\left|\left[\tilde{\mathcal{P}}_{k,0}^1 \otimes \cdots \otimes \tilde{\mathcal{P}}_{k,0}^{\ell-1} \otimes \tilde{\mathcal{P}}_k(\cdot|e) - \mathcal{P}_{k,0}^1 \otimes \cdots \mathcal{P}_{k,0}^{\ell-1} \otimes \mathcal{P}_k(\cdot|e)\right] V_{h+1}^k[k]^{\otimes \ell}\right| \leq b(k,h,\ell,e)$$

holds for every $k \geq 1, 1 \leq \ell \leq L, e \in \mathcal{E}_\ell$ with probability at least $1 - \delta$. $\qquad\square$

Next we present a lemma indicating that if the event in Lemma A.1 holds, then $V_h^k$ is always an optimistic estimation to the true value function $V_h^*$.

**Lemma A.2.** *Suppose the event in Lemma A.1 holds, then*

$$V_h^k[k] \geq V_h^*[k]$$

*holds for every $1 \leq h \leq H$ and $1 \leq k \leq K$.*

*Proof.* We will prove a stronger version of this lemma. That is, for every $(s^1, \cdots, s^L) \in \mathcal{S}_1 \times \cdots \times \mathcal{S}_L$, we have

$$V_h^k(s^1, \cdots, s^L) \geq V_h^*(s^1, \cdots, s^L).$$

We will prove this result by induction. When $h = H$, since $V_h^k = V_h^* = 0$, the inequality $V_h^k[k] \geq V_h^*[k]$ already holds. Next we assume that this inequality holds for $h+1$, and we consider the case $h$. For any $(s^1, \cdots, s^L) \in \mathcal{S}_1 \times \cdots \times \mathcal{S}_L, a^\ell \in \mathcal{A}_\ell$, we let $e = (s^\ell, a^\ell, s^{\ell-1})$. Suppose $\sigma(h) = \ell$, and for every $i < \ell$, $s^\ell$, $s^i$ are all end-state of level $i$. According to the events in Lemma A.1, we have

$$Q_h^k(s^1, \cdots, s^L; a^\ell) = r(s^1, \cdots, s^L, a_{h_\ell}^\ell) + \left[\tilde{\mathcal{P}}_{k,0}^1 \otimes \cdots \otimes \tilde{\mathcal{P}}_{k,0}^{\ell-1} \otimes \tilde{\mathcal{P}}_k(\cdot|e)\right] V_{h+1}^k(\cdots, \cdot, s^{l+1}, \cdots, s^L)$$

$$+ b(k,h,\ell,e)$$

$$\geq r(s^1, \cdots, s^L, a_{h_\ell}^\ell) + \left[\mathcal{P}_{k,0}^1 \otimes \cdots \otimes \mathcal{P}_{k,0}^{\ell-1} \otimes \mathcal{P}_k(\cdot|e)\right] V_{h+1}^k(\cdots, \cdot, s^{l+1}, \cdots, s^L)$$

$$\geq r(s^1, \cdots, s^L, a_{h_\ell}^\ell) + \left[\mathcal{P}_{k,0}^1 \otimes \cdots \otimes \mathcal{P}_{k,0}^{\ell-1} \otimes \mathcal{P}_k(\cdot|e)\right] V_{h+1}^*(\cdots, \cdot, s^{l+1}, \cdots, s^L)$$

$$= Q_h^*[k, a_{h_\ell}^\ell],$$

where the first inequality uses the event in Lemma A.1, and last inequality uses the fact that $V_{h+1}^k \geq V_{h+1}^*$ and $\left[ \mathcal{P}_{k,0}^1 \otimes \cdots \otimes \mathcal{P}_{k,0}^{\ell-1} \otimes \mathcal{P}_k(\cdot|e) \right]$ is a nonnegative operator. Therefore, we have

$$V_h^k(s^1, \cdots, s^L) = \min \left\{ \max_{a^\ell \in \mathcal{A}_\ell} Q_h^k(s^1, \cdots, s^L; a^\ell), H \right\}$$

$$\geq \min \left\{ \max_{a^\ell \in \mathcal{A}_\ell} Q_h^*(s^1, \cdots, s^L; a^\ell), H \right\} = V_h^*(s^1, \cdots, s^L)$$

This indicates that this lemma holds for $h$, which completes the induction. Hence for every $1 \leq h \leq H$, this lemma holds. $\square$

Equipped with these two lemma, we are ready to prove Theorem 4.1.

*Proof.* Suppose the event in Lemma A.1 always holds (which will happen with probability at least $1 - \delta$), then we can calculate the value function.

$$V_h^k[k] - V_h^{\pi_k}[k] = Q_h^k[k, a_{h_\ell}^\ell] - Q_h^{\pi_k}[k, a_{h_\ell}^\ell]$$

$$= \left[ \tilde{\mathcal{P}}_{k,0}^1 \otimes \cdots \otimes \tilde{\mathcal{P}}_{k,0}^{l-1} \otimes \tilde{\mathcal{P}}_k(\cdot|e_{k,l}) \right] V_{h+1}^k[k]^{\otimes \ell} + b(k, h, \ell, e_{k,h})$$

$$- \left[ \mathcal{P}_{k,0}^1 \otimes \cdots \otimes \mathcal{P}_{k,0}^{l-1} \otimes \mathcal{P}_k(\cdot|e_{k,l}) \right] V_{h+1}^{\pi_k}[k]^{\otimes \ell}$$

$$= \left[ \mathcal{P}_{k,0}^1 \otimes \cdots \otimes \mathcal{P}_{k,0}^{l-1} \otimes \mathcal{P}_k(\cdot|e_{k,l}) \right] \left( V_{h+1}^k[k]^{\otimes \ell} - V_{h+1}^{\pi_k}[k]^{\otimes \ell} \right) + b(k, h, \ell, e_{k,h})$$

$$+ \left[ \tilde{\mathcal{P}}_{k,0}^1 \otimes \cdots \otimes \tilde{\mathcal{P}}_{k,0}^{l-1} \otimes \tilde{\mathcal{P}}_k(\cdot|e_{k,l}) - \mathcal{P}_{k,0}^1 \otimes \cdots \mathcal{P}_{k,0}^{l-1} \otimes \mathcal{P}_k(\cdot|e_{k,l}) \right] V_{h+1}^k[k]^{\otimes \ell}$$

$$= \left( V_{h+1}^k[k] - V_{h+1}^{\pi_k}[k] \right) + \xi_{h+1,k} + b(k, h, \ell, e_{k,h})$$

$$+ \left[ \tilde{\mathcal{P}}_{k,0}^1 \otimes \cdots \otimes \tilde{\mathcal{P}}_{k,0}^{l-1} \otimes \tilde{\mathcal{P}}_k(\cdot|e_{k,l}) - \mathcal{P}_{k,0}^1 \otimes \cdots \mathcal{P}_{k,0}^{l-1} \otimes \mathcal{P}_k(\cdot|e_{k,l}) \right] V_{h+1}^k[k]^{\otimes \ell}$$

$$\leq \left( V_{h+1}^k[k] - V_{h+1}^{\pi_k}[k] \right) + \xi_{h+1,k} + 2b(k, h, \ell, e_{k,h}),$$

where the last inequality uses Lemma A.2, and we define $\xi_{h+1,k}$ as follows:

$$\xi_{h+1,k} = \left[ \mathcal{P}_{k,0}^1 \otimes \cdots \otimes \mathcal{P}_{k,0}^{l-1} \otimes \mathcal{P}_k(\cdot|e_{k,l}) \right] \left( V_{h+1}^k[k]^{\otimes \ell} - V_{h+1}^{\pi_k}[k]^{\otimes \ell} \right) - \left( V_{h+1}^k[k] - V_{h+1}^{\pi_k}[k] \right).$$

Sum up this inequality for all $1 \leq h \leq H$, and noticing that $V_H^k = V_H^{\pi_k} = 0$, we get

$$V_1^k[k] - V_1^{\pi_k}[k] \leq \sum_{h=1}^H \xi_{h+1,k} + 2 \sum_{h=1}^H b(k, h, \sigma(h), e_{k,h}),$$

which indicates that

$$\sum_{k=1}^K V_1^*[k] - V_1^{\pi_k}[k] \leq \sum_{k=1}^K V_1^k[k] - V_1^{\pi_k}[k] \leq \sum_{k=1}^K \sum_{h=1}^H \xi_{h+1,k} + 2 \sum_{k=1}^K \sum_{h=1}^H b(k, h, \sigma(h), e_{k,h}). \quad (11)$$

As for the first term in the above , it is easy to see that $\xi_{h+1,k}$ is a martingale difference sequence with respect to $h, k$. Since every $\xi_{h,k}$ is bounded by $H = H_1 \cdots H_L$, according to Azuma-Hoeffding inequality we have

$$\left| \sum_{k=1}^K \sum_{h=1}^H \xi_{h+1}^k \right| \leq 4H \sqrt{HK \log \delta^{-1}} \quad (12)$$

with probability at least $1 - \delta$.

Next, we will analyze the second term in equation 11. According to formula equation 10, we have

$$
\sum_{k=1}^{K}\sum_{h=1}^{H} b(k, h, \sigma(h), e_{k,h})
$$

$$
= H\sum_{k=1}^{K}\sum_{h=1}^{H}\min\left\{1, \sqrt{\frac{8(|\mathcal{S}_{\sigma(h)}| + \log(4L^2|\mathcal{S}_{\sigma(h)}||\mathcal{E}_{\sigma(h)}|k^2/\delta))}{n(k-1, e_{k,h})}}\right\} \tag{13}
$$

$$
+ H\sum_{k=1}^{K}\sum_{h=1}^{H}\sum_{i=1}^{\sigma(h)-1}\min\left\{1, \sqrt{\frac{8(|\mathcal{S}_i| + \log(4L^2|\mathcal{S}_i||\mathcal{E}_{\sigma(h)}|k^2/\delta))}{(k-1)\tilde{H}_i}}\right\}.
$$

In the first summation of the above equation, the one involved with $\sqrt{1/n(k-1, e_{k,h})}$ appears $n(k, e_{k,h}) - n(k-1, e_{k,h})$ times. And given $1 \le \ell \le L$, there exists $\tilde{H}_{\ell-1} - \tilde{H}_\ell$ choices of $h$ such that $\sigma(h) = \ell$. For given $\ell$, there are $|\mathcal{E}_\ell|$ choices of $n(k, e)$ where $e \in \mathcal{E}_\ell$. Therefore, we have

$$
\sum_{k=1}^{K}\sum_{h=1,\sigma(h)=\ell}^{H}\min\left\{1, \sqrt{\frac{8(|\mathcal{S}_\ell| + \log(4L^2|\mathcal{S}_\ell||\mathcal{E}_\ell|k^2/\delta))}{n(k-1, e_{k,h})}}\right\}
$$

$$
= \sum_{k=1}^{K}\sum_{e\in\mathcal{E}_\ell}(n(k,e) - n(k-1,e))\min\left\{1, \sqrt{\frac{8(|\mathcal{S}_\ell| + \log(4L^2|\mathcal{S}_\ell||\mathcal{E}_\ell|k^2/\delta))}{n(k-1, e)}}\right\}
$$

$$
= \sum_{e\in\mathcal{E}_\ell}\sum_{k=1,n(k-1,e)\le\tilde{H}_\ell}^{K}(n(k,e) - n(k-1,e))
$$

$$
+ \sum_{e\in\mathcal{E}_\ell}\sum_{k=1,n(k-1,e)>\tilde{H}_\ell}^{K}(n(k,e) - n(k-1,e))\sqrt{\frac{8(|\mathcal{S}_\ell| + \log(4L^2|\mathcal{S}_\ell||\mathcal{E}_\ell|k^2/\delta))}{n(k-1, e)}}
$$

$$
\le 2\tilde{H}_{\ell-1}|\mathcal{E}_\ell| + 2\cdot\sum_{k=1,n(k-1,e)>\tilde{H}_\ell}^{K}\sum_{e\in\mathcal{E}_\ell}(n(k,e) - n(k-1,e))\sqrt{\frac{8(|\mathcal{S}_\ell| + \log(4L^2|\mathcal{S}_\ell||\mathcal{E}_\ell|k^2/\delta))}{n(k, e)}}
$$

$$
\le \sum_{k=1}^{K}\sum_{e\in\mathcal{E}_\ell}\sum_{j=n(k-1,e)+1}^{n(k,e)}\sqrt{\frac{8(|\mathcal{S}_\ell| + \log(4L^2|\mathcal{S}_\ell||\mathcal{E}_\ell|k^2/\delta))}{j}}
$$

$$
= 2\tilde{H}_\ell|\mathcal{E}_\ell| + \sum_{e\in\mathcal{E}_\ell}\sum_{j=1}^{n(K,e)}\sqrt{\frac{8(|\mathcal{S}_\ell| + \log(4L^2|\mathcal{S}_\ell||\mathcal{E}_\ell|k^2/\delta))}{j}}
$$

$$
\le 2\tilde{H}_\ell|\mathcal{E}_\ell| + \sum_{e\in\mathcal{E}_\ell}2\sqrt{8(|\mathcal{S}_\ell| + \log(4L^2|\mathcal{S}_\ell||\mathcal{E}_\ell|k^2/\delta))}\cdot\sqrt{n(K,e)}
$$

$$
\le 2\tilde{H}_\ell|\mathcal{E}_\ell| + 2\sqrt{8(|\mathcal{S}_\ell| + \log(4L^2|\mathcal{S}_\ell||\mathcal{E}_\ell|k^2/\delta))}\cdot\sqrt{|\mathcal{E}_\ell|\cdot\sum_{e\in\mathcal{E}_\ell}n(K,e)}
$$

$$
= 2\tilde{H}_\ell|\mathcal{E}_\ell| + 2\sqrt{8(|\mathcal{S}_\ell| + \log(4L^2|\mathcal{S}_\ell||\mathcal{E}_\ell|k^2/\delta))}\cdot\sqrt{|\mathcal{E}_\ell|\cdot K(\tilde{H}_\ell - \tilde{H}_{\ell+1})}
$$

$$
= \tilde{O}\left(\tilde{H}_\ell|\mathcal{E}_\ell| + \sqrt{K\tilde{H}_\ell\cdot|\mathcal{E}_\ell|(|\mathcal{S}_\ell| + \log\delta^{-1})}\right),
$$

where the third inequality uses the fact that for any $e \in \mathcal{E}_\ell$ we have $n(k,e) - n(k-1,e) \le \tilde{H}_\ell$, and the second last equation uses the fact that $\sum_{e\in\mathcal{E}_\ell}n(K,e)$ is the number of all possible state-action-endstate pair appears up to $K$ horizons, which is $K$ times the number of $m$ such that $\sigma(h) = \ell$.

Therefore, the first term in equation 13 has the following estimation

$$H \sum_{k=1}^{K} \sum_{h=1}^{H} \min \left\{ 1, \sqrt{\frac{8(|\mathcal{S}_{\sigma(h)}| + \log(4L^2|\mathcal{S}_{\sigma(h)}||\mathcal{E}_{\sigma(h)}|k^2/\delta))}{n(k, e_{k,h})}} \right\}$$

$$= \sum_{\ell=1}^{L} \tilde{O} \left( H\tilde{H}_\ell |\mathcal{E}_\ell| + H\sqrt{K\tilde{H}_\ell \cdot |\mathcal{E}_\ell|(|\mathcal{S}_\ell| + \log \delta^{-1})} \right).$$

As for the second term in equation 13, we have

$$H \sum_{k=1}^{K} \sum_{h=1}^{H} \sum_{i=1}^{\sigma(h)-1} \min \left\{ 1, \sqrt{\frac{8(|\mathcal{S}_i| + \log(4L^2|\mathcal{S}_i||\mathcal{E}_{\sigma(h)}|k^2/\delta))}{(k-1)\tilde{H}_i}} \right\}$$

$$= H \sum_{k=1}^{K} \sum_{\ell=1}^{L} \sum_{h=1,\sigma(h)=\ell}^{H} \sum_{i=1}^{\ell-1} \min \left\{ 1, \sqrt{\frac{8(|\mathcal{S}_i| + \log(4L^2|\mathcal{S}_i||\mathcal{E}_\ell|k^2/\delta))}{(k-1)\tilde{H}_i}} \right\}$$

$$= H \sum_{k=1}^{K} \sum_{i=1}^{L} \sum_{\ell=i+1}^{L} \sum_{h=1,\sigma(h)=\ell}^{H} \min \left\{ 1, \sqrt{\frac{8(|\mathcal{S}_i| + \log(4L^2|\mathcal{S}_i||\mathcal{E}_\ell|k^2/\delta))}{(k-1)\tilde{H}_i}} \right\}$$

$$\leq H \sum_{k=1}^{K} \sum_{i=1}^{L} \sum_{\ell=i+1}^{L} \sum_{h=1,\sigma(h)=\ell}^{H} \min \left\{ 1, \sqrt{\frac{8(|\mathcal{S}_i| + \log(4L^2|\mathcal{S}_i|(\sum_{j=1}^{L} |\mathcal{E}_j|)k^2/\delta))}{(k-1)\tilde{H}_i}} \right\}$$

$$= H \sum_{k=1}^{K} \sum_{i=1}^{L} \tilde{H}_i \min \left\{ 1, \sqrt{\frac{8(|\mathcal{S}_i| + \log(4L^2|\mathcal{S}_i|(\sum_{j=1}^{L} |\mathcal{E}_j|)k^2/\delta))}{(k-1)\tilde{H}_i}} \right\}$$

$$\leq H \sum_{i=1}^{L} \tilde{H}_i + H \sum_{k=1}^{K} \sum_{i=1}^{L} \tilde{H}_i \sqrt{\frac{8(|\mathcal{S}_i| + \log(4L^2|\mathcal{S}_i|(\sum_{j=1}^{L} |\mathcal{E}_j|)k^2/\delta))}{(k-1)\tilde{H}_i}}$$

$$\leq \sum_{i=1}^{L} \tilde{O} \left( H\tilde{H}_i + H\sqrt{K\tilde{H}_i(|\mathcal{S}_i| + \log \delta^{-1})} \right),$$

where in the last inequality we apply the Hölder inequality. Combined previous two estimations together, we obtain that

$$\sum_{k=1}^{K} \sum_{h=1}^{H} b(k, h, \sigma(h), e_{k,h})$$

$$\leq \sum_{\ell=1}^{L} \tilde{O} \left( H\tilde{H}_\ell |\mathcal{E}_\ell| + H\sqrt{K\tilde{H}_\ell \cdot |\mathcal{E}_\ell|(|\mathcal{S}_\ell| + \log \delta^{-1})} \right)$$

$$+ \sum_{i=1}^{L} \tilde{O} \left( H\tilde{H}_i + H\sqrt{K\tilde{H}_i(|\mathcal{S}_i| + \log \delta^{-1})} \right)$$

$$= \sum_{\ell=1}^{L} \tilde{O} \left( H\sqrt{K\tilde{H}_\ell \cdot |\mathcal{E}_\ell|(|\mathcal{S}_\ell| + \log \delta^{-1})} \right)$$

This equation, together with equation 11 and equation 12, indicates the regret bound

$$R(K) = \sum_{k=1}^{K} V_1^*[k] - V_1^{\pi_k}[k]$$

$$\leq \sum_{\ell=1}^{L} \tilde{O} \left( H\tilde{H}_\ell |\mathcal{E}_\ell| + H\sqrt{K\tilde{H}_\ell \cdot |\mathcal{E}_\ell|(|\mathcal{S}_\ell| + \log \delta^{-1})} \right) + 4H\sqrt{HK \log \delta^{-1}}$$

$$= \sum_{\ell=1}^{L} \tilde{O} \left( H\tilde{H}_\ell |\mathcal{E}_\ell| + H\sqrt{K\tilde{H}_\ell \cdot |\mathcal{E}_\ell|(|\mathcal{S}_\ell| + \log \delta^{-1})} \right)$$

holds with probability at least $1 - 2\delta$. This completes the proof of Theorem 4.1. $\qquad\square$

