# OpenReview forum: "PROVABLY BENEFITS OF DEEP HIERARCHICAL RL"
_ICLR.cc/2020/Conference — Reject_

### Official Review · AnonReviewer4 · 2019-10-18
**Official Blind Review #4**

**Rating:** 3

**Review:**

This paper performs a regret analysis for a new hierarchical reinforcement learning (HRL) algorithm that claims an exponential improvement over applying a naive RL approach to the same problem. The proposed algorithm and the regret analysis performed seem rigorous and well-thought out.

However, I think that this paper should be rejected because (1) the algorithm does not appear to be a substantial improvement over existing algorithms, (2) the paper makes strong claims about an exponential improvement over standard RL, but doesn't provide a strong benchmark to compare to, and (3) the paper is imprecise and unpolished, with many grammatical errors.

I would be open to reconsidering my score if a) the authors submit a revised version with significantly cleaned up text, and b) if the authors could provide more information about how their contribution compares to the existing literature.

Main argument

The paper would benefit from establishing stronger context for the central contributions of their paper. For instance, the paper begins by contrasting HRL approaches with a number of standard RL algorithms, saying that approaches such as AlphaGo do not require high-level planning. This seems surprising; many RL researchers would describe MCTS (the base of the AlphaGo algorithm) as performing planning. It would be great if the authors could go into more detail as to what they view as planning, and why AlphaGo does not do so.

Additionally, the main comparison the authors seem to make is between HRL and naive RL, which does not provide sufficient context to properly analyse their algorithm. Many algorithms are better than applying a classical RL algorithm naively. As such, it is not sufficient to show that the algorithm proposed by the authors is stronger than a naive approach; it would be better to compare the algorithm to either a) the state of the art (SOTA) approach, or b) a more credible approach than the naive one. Experimental evidence would help.

One point of comparison is Fruit et al. (2017), which is mentioned as another paper which carries out a regret analysis in a HRL setting. Fruit et al. (2017) contains a number of simple numerical simulations; a similar effort here would help.

Another issue is that the paper is confusing, with systematic grammar errors and typos. The paper would benefit significantly with some copy-editing/proofreading by a native English speaker. For instance, the title should (presumably) read "Provable Benefits of Deep Hierarchical RL." Such errors appear throughout the paper. Fixing them would make the paper much easier to understand.

Finally, although this did not factor into the score I awarded the paper, the terminology used by the authors is confusing, referring to their setting as "Deep Hierarchical Reinforcement Learning." "Deep Reinforcement Learning" is a widely used term in industry, referring to algorithms that apply Deep Learning to RL problems, such as AlphaGo or DeepStack. I would encourage the authors to use a different term to describe the setting.

Questions to the authors:

1) In what way is AlphaGo not doing planning? What is an example of an algorithm that does planning in a standard RL setting? e.g. what would planning look like in Go?
2) Did you run any experiments/simulations of your work? If not, why not?
3) Can you elaborate on what a classical RL algorithm would look like that would serve as a proper benchmark to this algorithm?
4) In your mind, what is the SOTA algorithm for your setting?
5) What are some simple domains that your algorithm would apply to?

[0]: Moravčík, Matej & Schmid, Martin & Burch, Neil & Lisý, Viliam & Morrill, Dustin & Bard, Nolan & Davis, Trevor & Waugh, Kevin & Johanson, Michael & Bowling, Michael. (2017). DeepStack: Expert-Level Artificial Intelligence in No-Limit Poker. Science. 356. 10.1126/science.aam6960.

**Experience Assessment:**

I have published one or two papers in this area.

**Review Assessment: Checking Correctness Of Derivations And Theory:**

I assessed the sensibility of the derivations and theory.

**Review Assessment: Checking Correctness Of Experiments:**

N/A

**Review Assessment: Thoroughness In Paper Reading:**

I read the paper thoroughly.

---

> ### Author Response · Authors · 2019-11-15
> **Response**
>
> Thank you for the questions and suggestions.
> We want to emphasize that our goal is to formalize the deep hierarchical reinforcement learning problem and give a provably efficient algorithm for this setting. The main focus is theoretical, and we do not claim to beat any SOTA algorithm.

---

### Official Review · AnonReviewer3 · 2019-10-23
**Official Blind Review #3**

**Rating:** 1

**Review:**

This paper proposes a new kind of episodic finite MDPs called "deep hierarchical MDP" (hMDP). An L-layer hMDP can be *roughly* thought of as L episodic finite MDPs stacked together. A variant of UCRL2 [JOA10] is proposed to solve these hMDPs and some results from its regret analysis are provided.

Pros:

1. The essential result (Theorem 4.1) on the regret bound of the proposed algorithm seems correct. I have not checked the proofs in detail but in part because it does not seem surprising and that a precise assessment is hindered by many typos (see Min2 and Con2).

Cons (in descending order of their weights in my decisions):

1. The proposed hMDPs do _not_ seem to capture important features or challenges in hierarchical RL. My understanding is that the transitions in hMDPs work _like_ a clockwork (more on this in Mis6), the algorithm interacts with the sub-MDPs at each layer in turns according to their fixed horizons H_l's. This structure is very rigid temporally and seems to exclude the mentioned example of autonomous driving: the number of decision steps between intersections would be fixed.

2. There are many (typographical) errors in both the text and mathematical expressions. Some of them are more severe than others hindering understanding.

3. Possible as a consequence of Con2, some quantities defined seem unclear or incorrect at worst. For example, the "standard regret" defined in (2) is an expectation, not a random variable as in convention.

4. There are some notable deviations from similar settings in prior works. They might be worthwhile innovations but their significance or motivations is omitted. For example, the rewards in hMDPs are defined as a function of the full state, i.e. in general not decomposable to rewards on the states of each layer, yet the analogy for hMDP is "L levels of episodic MDPs."

A non-exhaustive list of obvious mistakes/typos:
1. In the title, "Provably" -> Provable.
2. In the abstract, "often both" -> often requires both.
3. In Organization, "theoremm" -> theorems.
4. In Section 2, “between exploration” -> between exploration and exploitation.
5. Above Section 3, "carried" -> carried out.
6. Below (1), "amount reward" -> amount of reward.
7. The definition of horizon H is incorrect. Consider H_1 = 2 and H_2 = 3, the algorithm will interact with the sub-MDPs in the following order within one episode: 1, 1, 2, 1, 1, 2, 1, 1, 2. There are 9 steps not 6 = 2 * 3 as defined.
8. Section 3.3, "able accumulate" -> able to accumulate.
9. Section 3.3, the definition of V_h^\pi, there should be not \max.
10. (5), "H" -> H - h.
11. Section 6, "tabular R" -> tabular RL.
12. In References, "Posterior sampling for reinforcement learning: worst-case regret bounds" -> Optimistic posterior sampling for reinforcement learning: worst-case regret bounds.
13. In References, "Temporal abstraction in reinforcement learning" should be cited as a PhD thesis.

Some other possible errors/inconsistencies:
1. Related work listed regret bounds from prior works (the presentation closely mirrors that of [JABJ18]) assume an episodic MDP with non-stationary transitions, i.e. P_t ≠ P_{t'} in general. However, in 3.1 the transitions are stationary. Relatedly, regardless of the stationarity of the transitions, there may not be an optimal _stationary_ policy in an episodic MDP contrary to the claim in the paper.
2. Indexing seems inconsistent near the top of page 3. The initial state is s_0 but the trajectory starts with s_1.
3. Near the top of page 3, V_h^\pi and Q_h^\pi should sum from h'=h, not h'=1. I assume that the authors intend to define h-step values (to appear in the Bellman equations).
4. Section 3.3, what are the k's in the equations?
5. (6), what is n(k-1, e)?

Minor (factored little to none in my decision):
1. The claim in Introduction that some games "do not require high-level planning" while others do is highly speculative and vague. Note that any policy can be written a function with codomain in the primitive actions. In fact, many people thought to solve a game like chess or Go requires some temporal hierarchy (opening, mid-game, and end-game).
2. The comparison to running UCRL2 on hMDP ignoring the given structure seems weak. Given the knowledge of the particular clockwork-like structure of hMDP at each layer (horizons, states, actions), the natural attempt would be run O(L) copies of UCRL2, one for each sub-MDP (under different terminating states of the immediately lower sub-MDP). Frankly, in my understanding, that seems to be roughly what the authors propose as the solution (thus the results unsurprising). Moreover, it is not immediately clear that UCRL2 can apply to the proposed setting of hMDP without checking regular conditions like communicating (diameter being finite).
3. The claim that RL with options “can be viewed as a two-layer HRL” needs much elaboration if not correction. Note that in the former, primitive actions are always taken in the original MDP at consecutive steps.
4. There is a limited relevance to deep learning or deep RL central to the themes at ICLR, i.e. the general issue of representation. This work may be more suitable for other general ML venues.

Some suggestions

I agree with the authors' sentiment that our theoretical understanding of hierarchical RL is relatively limited. I applaud the authors' effort to address this limitation. But judging from this aim of advancing our theoretical understanding, I think the paper may be improved by

1. better articulating the motivations for hMDPs (concrete examples would help)

2. contextualizing hMDPs with respect to other well-known models such as semi-MDPs (technical and precise comparison would help).

To put it in a different way, it is unclear to the readers why we want to solve this special class of hMDPs and what does hMDPs have to do with the general issues in hierarchical RL. Technically, I feel that assuming episodicity seems against the spirit of hierarchical RL where subtasks are often delimited by their subgoals instead of durations.

In conclusion, I cannot recommend accepting the current article.

(To authors and other reviewers) Please do not hesitate to directly point out my misunderstandings if there is any. I am open to acknowledging mistakes and revising my assessment accordingly.


Post-rebuttal update:

Thank you for replying to my review and incorporating some of my suggestions into your revision. However, I found many concerns (and mistakes) unaddressed, such as Mis7. The use of driving in Manhattan as an example troubles me because even stopping for a traffic light seems to disrupt the fixed temporal hierarchy of decisions. In conclusion, I will maintain my recommendation.

**Experience Assessment:**

I have published one or two papers in this area.

**Review Assessment: Checking Correctness Of Derivations And Theory:**

I assessed the sensibility of the derivations and theory.

**Review Assessment: Checking Correctness Of Experiments:**

N/A

**Review Assessment: Thoroughness In Paper Reading:**

I read the paper thoroughly.

---

> ### Author Response · Authors · 2019-11-15
> **Response**
>
> Thank you for the questions and suggestions. We have revised our paper and fixed typos. Please find out responses to your comments below.
> 1. For autonomous driving, if we assume that each road has the same lengths, and our vehicle needs to make a decision after going a certain distance, then indeed the number of decision steps between interactions is fixed. When the lengths of streets are of the same lengths, as long as they are straight like roads in Manhattan, our model is also suitable after slight modification.
> 2. The episodic way given in our model is a different explanation of the hierarchical model in comparison to models like option MDP, where jumps between layers happen when meeting the stopping criterion. Our model is more suitable for a situation like autonomous driving, or some computer games where we have a time limit in each challenge since in these cases, the number of steps in each layer is fixed.
> 3. The “deep” in our model means deep layers of hierarchy, instead of algorithms using deep learning or deep RL.

---

### Official Review · AnonReviewer1 · 2019-10-25
**Official Blind Review #2**

**Rating:** 3

**Review:**

This paper studies the theoretical aspects of HRL. It provides theoretical analysis for the complexity of Deep HRL. The idea is to exploit a given action hierarchy, and known state decomposition, the fact that the high-level state space shares similar low-level structures. The final result is an exponential improvement of HRL to flat RL.

Overall, the paper pursues an ambitious goal that analyses the complexity of Deep HRL. The writing is not easy to follow. I some questions and concerns as follows

- I wonder why the state space must be defined in a product form? If a standard RL is used, then it could be applied directly to the state space ($S_L$) on that primitive actions operate. Hence L-1 state spaces will be discarded? I don't see why a flat RL must estimate policies for states at all levels. It looks like many later derivations based on the assumption of factored state spaces and factored transitions on different levels. In the case of factored representation, the authors should make clear assumptions and find a better way to describe the overall algorithm.

- Section 3.2: the authors use time index for Q and V, does that mean all analysis is for non-stationary MDPs? This is not the assumption in Jaksch et al. (2010) and this paper. The description in this section is very confusing and contains a lot of imprecise definitions
e.g. should H = \prod {i=1} H_i?? is h =(h_1,...h_L) not in [1,H]? what is the definition of the immediate next lexicographical tuple? etc. The definition of \sigma is also unclear and hard to understand.

- The analysis in Section 4. and Algorithm 1 are not for Deep HRL as said in Abstract and Introduction. The analysis is based on PAC-MDP learning for models at each action level. This paper's contributions might be clearer if the authors made clearer assumptions, e.g. on action hierarchy, abstract state space structures etc..



**Experience Assessment:**

I have published one or two papers in this area.

**Review Assessment: Checking Correctness Of Derivations And Theory:**

I assessed the sensibility of the derivations and theory.

**Review Assessment: Checking Correctness Of Experiments:**

I assessed the sensibility of the experiments.

**Review Assessment: Thoroughness In Paper Reading:**

I read the paper at least twice and used my best judgement in assessing the paper.

---

> ### Author Response · Authors · 2019-11-15
> **Response**
>
> Thanks for these questions and suggestions. We have revised our paper and fixed typos. Please find our responses to your questions below.
> 1. In our model, we assume the transition model shares the hierarchical structure, but the reward can be arbitrary (the reward is defined as $r(s_1, s_2, …, s_L, a)$, which is a function of states at every level and the action). Hence we have to plan on the product state space and make actions for all states at all levels.
> 2. Here we use indexed Q and V to denote the value functions at different horizons, which are common notations for finite-horizon MDP. As for the notation $h=(h_1, …, h_L)\in [1, H]$, we actually means $h=h_L+h_{L-1}*H_L+…+h_1*H_2*H_3*…*H_L$, which is the lexicographical number of tuple $(h_1, …, h_L)$. The lexicographical tuple next to $h = (h_1, …, h_L)$ is $(h_1, …, h_l+1, 1, 1, …, 1)$ if $l$ is the largest index such that $h_l < H_l$ (meaning tuple $h=(h_1, …, h_l, H_{l+1}, ..., H_L)$). Also, in the previous tuple, we use $l=\sigma(h)$ to denote the level where the carry happens.
> 3. In our model, we use “deep hierarchical RL“ to denote the model with many layers requiring planning.

---

### Decision · Program_Chairs · 2019-12-19

**Decision:**

Reject

**Comment:**

This paper pursues an ambitious goal to provide a theoretical analysis HRL in terms of regret bounds. However, the exposition of the ideas has severe clarity issues and the assumptions about HMDPs used are overly simplistic to have an impact in RL research.
Finally, there is agreement between the reviewers and AC that the novelty of the proposed ideas is a weak factor and that the paper needs substantial revision.